# Lanthanide-regulating Ru-O covalency optimizes acidic oxygen evolution electrocatalysis

Lu Li[1,3], Gengwei Zhang[2,3], Chenhui Zhou[1], Fan Lv[1], Yingjun Tan[1], Ying Han[1], Heng Luo[1], Dawei Wang[1], Youxing Liu[1], Changshuai Shang[1], Lingyou Zeng[1], Qizheng Huang[1], Ruijin Zeng[1], Na Ye[1], Mingchuan Luo[1] & Shaojun Guo[1] ✉

Precisely modulating the Ru-O covalency in $RuO_x$ for enhanced stability in proton exchange membrane water electrolysis is highly desired. However, transition metals with $d$-valence electrons, which were doped into or alloyed with $RuO_x$, are inherently susceptible to the influence of coordination environment, making it challenging to modulate the Ru-O covalency in a precise and continuous manner. Here, we first deduce that the introduction of lanthanide with gradually changing electronic configurations can continuously modulate the Ru-O covalency owing to the shielding effect of $5s/5p$ orbitals. Theoretical calculations confirm that the durability of $Ln-RuO_x$ following a volcanic trend as a function of Ru-O covalency. Among various $Ln-RuO_x$, $Er-RuO_x$ is identified as the optimal catalyst and possesses a stability 35.5 times higher than that of $RuO_2$. Particularly, the $Er-RuO_x$-based device requires only 1.837 V to reach $3\,A\,cm^{-2}$ and shows a long-term stability at $500\,mA\,cm^{-2}$ for 100 h with a degradation rate of mere $37\,\mu V\,h^{-1}$.

Renewable-driven water electrolysis is widely recognized as a promising and sustainable route to scalable production of green hydrogen[1–4]. Proton exchange membrane water electrolysis (PEMWE) technology shows great potential on account of its high intermittent compatibility, low Ohmic resistance, high current density, low operating pressure, and limited side reactions[5–7]. However, the widespread application of PEMWE is obstructed by the lack of efficient and cost-affordable electrocatalysts for acidic oxygen evolution reaction (OER)[8,9]. OER with four proton-coupled electron transfer necessitates a high overpotential due to its sluggish kinetics, thereby decreasing the operating efficiency of PEMWE[10,11]. Besides, strong oxidative bias and extremely acidic corrosion collectively challenge the durability of available OER electrocatalysts for high-current-density operation[12].

Currently, iridium (Ir)-based catalysts, e.g. $IrO_2$, remain the only practical choice for anode electrocatalysts in PEMWE due to their well balance in durability and activity[13,14]. However, the spreading of PEMWE is drastically limited by the availability of Ir, which eventually leads to severe Ir shortage[15,16]. In this context, numerous efforts have

been devoted to searching for alternatives to Ir, of which the most promising one is identified to be Ru-based oxides for their intrinsic high activity. Nevertheless, Ru-based OER catalysts always suffer from insufficient stability at industrial current density, which hampers their deployment in PEMWE for green hydrogen industry[17]. The instability of Ru-based anode originates from the over-oxidation of Ru species, generating soluble $RuO_4$ under OER potentials[18–20]. Despite considerable efforts on stabilizing Ru for OER electrocatalysis, a significant performance gap remains to meet the industrial requirements[21–23]. To fill this gap, it would be more efficient to establish a fundamental and manageable strategy for stabilizing Ru-based OER electrocatalysts, instead of the conventional trial-and-error approach.

The stability of nanocrystalline $RuO_2$-based catalysts is closely tied to the covalency of Ru−O bonds[6]. Weakening the Ru−O bond covalency can localize O $2p$ and Ru $3d$ orbitals below Fermi level, inhibiting lattice oxygen's participation in OER and the formation of oxygen vacancies, thereby preventing excessive overoxidation of Ru species into soluble $RuO_4$ during OER[24]. Meanwhile, excessively low Ru−O

[1]School of Materials Science and Engineering, Peking University, Beijing, China. [2]Faculty of Metallurgical and Energy Engineering, Kunming University of Science and Technology, Kunming, Yunnan, China. [3]These authors contributed equally: Lu Li, Gengwei Zhang. ✉e-mail: guosj@pku.edu.cn

covalency is detrimental as it makes Ru be easily leached, leading to the direct demetallation of surface Ru and the subsequent structural degradation of $RuO_2$[23]. Prior studies focused on modulating the electronic structure of $RuO_2$ using $3d$, $4d$, and $5d$ metal substrates/dopants[6,21,25]. However, as $d$ orbital locates the outermost of transition metals, it is susceptible to external influences of crystal field and coordination environment[26–28]. For instance, the introduction of consecutive elements such as Ni ($3d^84s^2$), Cu ($3d^{10}4s^1$), and Zn ($3d^{10}4s^2$) into $RuO_x$ results in the transformation of their valence electrons into $3d^8$, $3d^{10}$, and $3d^{10}$, respectively[20,29,30]. Such external influences render it be extremely challenging to modulate the Ru–O covalency in a precise and continuous manner.

Herein, we reason that lanthanide (Ln)-group elements with the $4f$ orbital buried under $5s/p$ can minimize external influences and consequently enable precise and continuous tuning of Ru–O covalency for durable OER electrocatalysis. Density functional theory (DFT) calculations were conducted on Ln-$RuO_x$ systems, showing that the resultant $4d–2p–4f$ hybridization induces a continuously varying Ru–O covalency for dictating OER performance. Benefiting from an optimal Ru–O covalency, Er-$RuO_x$ was screened out as it demonstrated the largest formation energy of the lattice oxygen and Ru vacancy. The operando characterizations confirm the critical role of Er dopants in stabilizing the Ru–O structure for improved OER durability. Moreover, the up-shifted $d_{z^2}$-state energy level (−0.855 eV) of Er-$RuO_x$, relevant to the benchmark $RuO_2$ (−2.171 eV), results in less electronic occupancy in the antibonding states and a stronger *OH adsorption, as validated by methanol molecular probe experiments, thereby significantly boosting the catalytic activity of Er-$RuO_x$. This work validates a novel Ln-regulating approach for precisely and continuously modulating Ru–O covalency, aiding to more economic affordability of PEMWE in a hydrogen economy scenario.

## Results

### Lanthanide-regulating Ru–O covalency as OER descriptor

Ln elements offer a flexible avenue for fine-tuning and optimizing the electrocatalytic performance of catalysts due to the shielding effect of $5s/5p$ orbitals, gradient-filled $4f$ orbital electron configuration, rich electronic energy levels, and the ability to accommodate various coordination numbers. The gradient orbital coupling of Ru, O, and Ln shows promise for enhancing the OER performance based on group theory-directed symmetric analysis[31,32]. For [$RuO_6$], the orbital coupling of valence $4d$, $5s$, and $5p$ orbitals with $sp$-mixing orbitals of coordinated O atoms produces metal-oxygen (M-O) bonding states composed of $a_{1g}$, $t_{1u}$, $e_g$, with their corresponding (M-O)* antibonding states $a_{1g}^*$, $t_{1u}^*$, $e_g^*$, and $t_{2g}$ non-bonding terms. For [$LnO_6$], the orbital coupling between Ln-$4f$ and O-$2p$ is contributed by $t_{1u}$ and $t_{2u}$ with their antibonding $t_{1u}^*$ and $t_{2u}^*$ terms, and $a_{2u}$ non-bonding terms. Thanks to the $\sigma$ conjunction with O-$p$ orbitals, the gradient orbital coupling of Ru–O–Ln can be formed (Fig. 1a), resulting in more flexible electronic interactions for electrocatalytic adaptation.

We first carried out the DFT calculations with Ru–O–Ln configuration constructed based on the rutile $RuO_2(110)$ (Supplementary Figs. 1–4). The Bader charge analysis shows electron transfers from Ln to Ru, reducing the charge of Ru sequentially from 1.52 ($RuO_2$) to 1.50 (Tm-$RuO_x$), 1.48 (Er-$RuO_x$), and 1.22 (Ho-$RuO_x$) (Supplementary Fig. 5), which confirms the $d$-$p$-$f$ orbital hybridization in Ln-$RuO_x$. Through the analysis of bonding and antibonding orbital filling, the crystal orbital Hamilton population (COHP) and integrated COHP (ICOHP) calculations results demonstrate that the introduction of Ln can weaken the Ru–O bonding state occupancy from −1.614 eV ($RuO_2$) to −1.523 eV (Ho-$RuO_x$), −1.573 eV (Er-$RuO_x$), and −1.574 eV (Tm-$RuO_x$) (Supplementary Fig. 6). The fine tuning of Ru–O covalency originates from the shielding effect of $5s/5p$ orbitals for Ln elements with gradient-filled $4f$ orbital electron configuration.

To evaluate the stability of lattice oxygen and Ru, we calculated the formation energy of the lattice oxygen ($\Delta G_{O\ vacancy}$) and Ru vacancy ($\Delta G_{Ru\ vacancy}$), which are utilized together to assess the stability of the electrocatalysts. As Ln dissolution would take place during OER process[33], defective structures containing Ln vacancies were constructed. In the presence of Ln vacancies, Er-$RuO_x$ exhibits the highest $\Delta G_{O\ vacancy}$ (0.33 eV), surpassing $RuO_2$, Ho-$RuO_x$, and Tm-$RuO_x$ by 0.29, 0.02, and 0.12 eV, respectively (Fig. 1b). Specifically, the regulation of Ru–O covalency leads to a modified $\Delta G_{Ru\ vacancy}$, increasing from 2.58 eV in $RuO_2$ to 3.49, 3.78, and 3.44 eV for Ho-$RuO_x$, Er-$RuOx$, and Tm-$RuO_x$, respectively (Fig. 1c). Considering the $\Delta G_{O\ vacancy}$ and $\Delta G_{Ru\ vacancy}$ in combination, the stability of Ln-$RuO_x$ follows the volcanic-like trend as a function of Ru–O covalency. This in turn verifies our proposed design principle that the fine tuning of Ru–O covalency by Ln regulates stability.

To further investigate the mechanism by which Ln regulated the OER activity, we calculated the Gibbs free energy of oxygen intermediates during OER, yielding the theoretical overpotential for Ho-$RuO_x$, Er-$RuO_x$, Tm-$RuO_x$, and $RuO_2$. As shown in Fig. 1d and Supplementary Figs. 7 and 8, the potential determining step (PDS) for $RuO_2$ is the evolution from *O to *OOH, with a calculated overpotential of 0.53 V, while the overpotential decreases to 0.48, 0.46, and 0.50 V for Ho-$ErO_x$, Er-$RuO_x$, and Tm-$RuO_x$, respectively. The theoretical overpotential of various electrocatalysts are depicted in Fig. 1e employing a three-dimensional volcano-shaped plot that delineates the free energy difference between *O and *OH intermediates. Benefitting from the optimized Ru–O covalency, Er-$RuO_x$ demonstrates an enhanced *OH-binding strength (Supplementary Fig. 9) and provides near-optimal free energies for each intermediate, thus leading to the low theoretical OER overpotential. For the samples with both lower (Ho-$RuO_x$, Tm-$RuO_x$) and higher ($RuO_2$) Ru–O covalency, the free energy increase of PDS is observed. Ru–O covalency of the catalysts reflects the Ru-*OH bonding interaction, suggesting an optimal Ru–O covalency that is neither too weak nor too strong is favorable for OER.

Overall, the incorporation of Ln can continuously optimize the Ru–O covalency within a narrow range, thereby controlling the dissolution kinetics of $RuO_2$-based catalysts and influencing the energy barriers of key reaction steps.

### Materials synthesis and characterization

As Er-$RuO_x$ theoretically possesses high OER activity and stability, it is selected for further investigation. To experimentally verify the predictions above, the Er-$RuO_x$ catalyst was prepared by a one-pot glucose-blowing method. Pure $RuO_2$ was also synthesized employing the same method for comparison. The scanning electron microscopy (SEM) images (Supplementary Fig. 10) manifest that Er-$RuO_x$ possesses a porous sheet-like structure with a specific surface area ($S_{BET}$) of 64.78 $m^2\ g^{-1}$ (Supplementary Fig. 11). Furthermore, the porous sheets are composed of a large number of small nanoparticles (Supplementary Fig. 12). The X-ray diffraction (XRD) patterns reveal that Er-$RuO_x$ possesses a rutile-type structure with no distinct diffraction peaks related to $ErO_x$ (Supplementary Fig. 13), suggesting the successful incorporation of Er atoms into $RuO_2$. High-resolution transmission electron microscopy (HRTEM) image shows Er-$RuO_x$ nanosheets consist of small nanoparticles with sizes around 5–10 nm, which are composed of the (110) facet-dominated $RuO_2$ (Supplementary Fig. 14).

Furthermore, we utilized the aberration-corrected transmission electron microscopy (AC-TEM) to reveal the fine atomic-scale structure of the Er-$RuO_x$ catalyst. The high-angle annular dark-field scanning transmission electron microscopy (HAADF-STEM) images (Fig. 2a, d) show the clear edge of the Er-$RuO_x$ grain. Figure 2b and Supplementary Fig. 15 display the surface plots of Er-$RuO_x$, allowing the depiction of the intensity of the atomic columns. The atomic configuration of Er-$RuO_x$ can be clearly seen from the atomic HAADF-STEM images along the [001] and [110] zone axes (Fig. 2c, e), consistent with that of pure

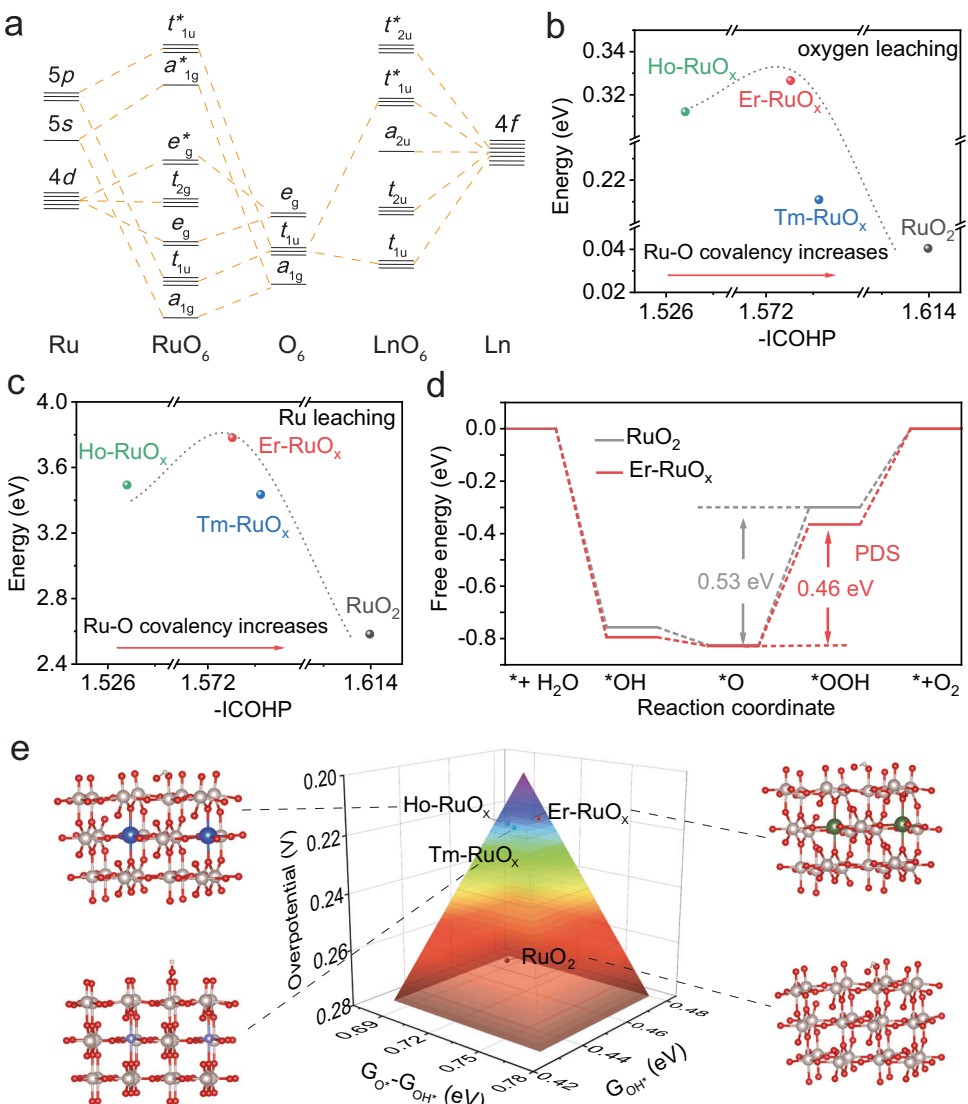

**Fig. 1 | Prediction of the OER performance utilizing DFT calculations.**
**a** The qualitative molecular orbital diagram obtained from [RuO$_6$] and [LnO$_6$].
**b** The $\Delta G_{O\ vacancy}$ and (**c**) $\Delta G_{Ru\ vacancy}$ as a function of -ICOHP for Ln-RuO$_x$. The upshift values of ICOHP indicate lower Ru–O covalency. **d** The reaction paths on Er-RuO$_x$ and RuO$_2$ at 1.23 V. **e** Volcano plot for different electrocatalysts and corresponding structures.

RuO$_2$. The energy-dispersive spectra (EDS) line scan in Fig. 2f was recorded from the HAADF-STEM image of Er-RuO$_x$ (Fig. 2d, the green line OO'). The O, Ru, and Er signals are detected and recorded in the line scan profile. Especially, the signals resulting from Er only appears in the regions of Ru signal, further indicating that Er was successfully introduced into RuO$_2$. The relatively weak signal intensity of Er element in contrast to that of Ru reveals the low Er content, matching with the result of EDS spectra (Supplementary Fig. 16). The element mapping images further demonstrate the coexistence and atomic-scale distribution of Ru and Er atoms (Fig. 2g–q). Er atoms are distributed within the catalyst, with a preference for surface localization, which contributes to the electrocatalytic performance as the Ru–O covalency can be regulated as discussed above.

### Electrocatalytic performance in three-electrode configurations and PEMWE devices

The OER performance of Er-RuO$_x$ and the control samples were measured in 0.5 M H$_2$SO$_4$ solution in a three-electrode system. As exhibited in Fig. 3a, Er-RuO$_x$ represents superior OER activity to commercial and home-made RuO$_2$. Excitingly, the required overpotential to reach current density of 10 mA cm$^{-2}$ is only 200 ± 8 mV on Er-RuO$_x$, 87 ± 5 and 77 ± 7 mV lower than those of commercial RuO$_2$ and home-made RuO$_2$, respectively. Furthermore, when normalized to the electrochemical active area, the catalytic activity of Er-RuO$_x$ remains better than that of commercial RuO$_2$ (Supplementary Fig. 19). The Er-RuO$_x$ catalyst also demonstrates a decreased Tafel slope of 45 mV dec$^{-1}$ compared to commercial RuO$_2$ (105 mV dec$^{-1}$) (Fig. 3b), suggesting the boosted reaction kinetics.

The stability of the as-prepared catalysts was investigated by cyclic voltammetry (CV) between 1 V and 1.45 V vs. RHE. Er-RuO$_x$ demonstrates a much smaller attenuation than commercial RuO$_2$ after 30,000 cycling tests (Fig. 3c). The concentration of dissolved Ru after 30,000 CV cycles for Er-RuO$_x$ was measured to be 13.7 ppb, which was much lower than that of commercial RuO$_2$ (40.9 ppb). These results suggest that the incorporation of Er suppresses the dissolution of RuO$_x$. In addition, the catalytic durability was evaluated by chronopotentiometry (CP) at 10 mA cm$^{-2}$ (Fig. 3d), demonstrating a more pronounced stability advantage of Er-RuO$_x$ over commercial RuO$_2$. In detail, the required overpotential of commercial RuO$_2$ increased by 674 mV after 73 h stability test at 10 mA cm$^{-2}$, which was ~35.5 times higher than that of Er-RuO$_x$, verifying the beneficial role of Er on catalytic stability. The OER performance of Ho-RuO$_x$ and Tm-RuO$_x$ was

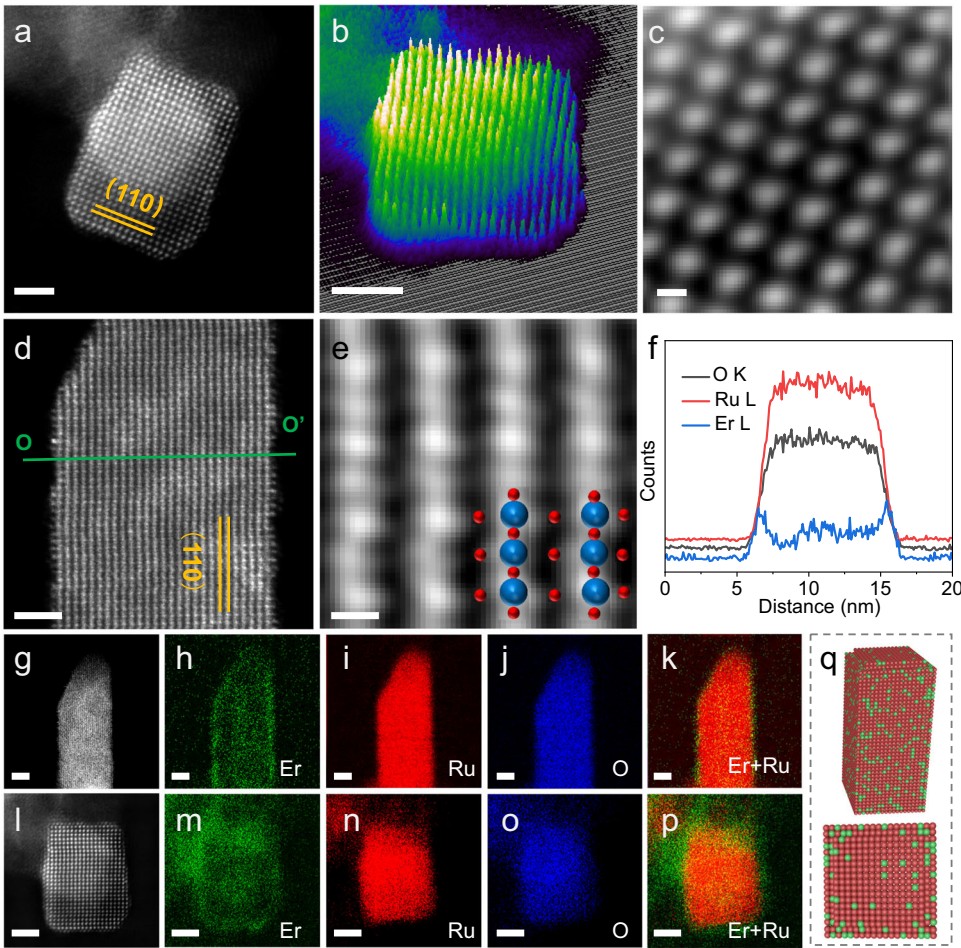

**Fig. 2 | Atomic-scale structure characterization of Er-RuOₓ. a** HAADF-STEM and **b** surface plot of Er-RuOₓ along [001] zone axes, scale bar: 2 nm. **c** Atomic STEM image along [001] zone axes and structural model (inset) of Er-RuOₓ, scale bar: 2 Å. **d** HAADF-STEM (scale bar: 2 nm) and (**e**) atomic STEM images (scale bar: 2 Å) of Er-RuOₓ along [110] zone axes. **f** EDS line scan of O, Ru, and Er signal recorded from the green line OO' in (**d**). HAADF-STEM image and corresponding element mapping images of Er-RuOₓ along the (**g–k**) [110] and (**l–p**) [001] zone axes, scale bar: 2 nm. **q** Schematic illustration of Er-RuOₓ.

shown in Supplementary Figs. 22–24, being in good agreement with DFT prediction. As shown in Supplementary Tables 6 and 7, both the activity and stability of Er-RuOₓ were higher than those of the recently reported Ru-based electrocatalysts[7,22,23,25], confirming the economic efficiency of Er-RuOₓ.

To investigate the application potential of Er-RuOₓ for water electrolysis, we constructed a PEMWE device (Nafion 117 membrane) using Er-RuOₓ and commercial Pt/C as the anode and cathode catalyst, respectively (Fig. 3e). The current-voltage curves (without *iR* compensation) in Fig. 3f clearly demonstrate the superior water electrolysis performance of the Er-RuOₓ-based PEMWE in comparison to the commercial RuO₂||Pt/C PEMWE device. Specifically, the Er-RuOₓ-based electrolyzer (at 80 °C) required only 1.590, 1.713, and 1.837 V to reach an industrial current density of 1, 2, and 3 A cm⁻², respectively, outperforming the PEMWE using the other state-of-the-art RuOₓ-based catalysts (Supplementary Table 8). Moreover, the PEMWE employing the Er-RuOₓ catalyst achieves an efficiency of approximately 80% at 1 A cm⁻² (Supplementary Fig. 26), with an estimated cost of about US\$ 0.85 *per* kg of H₂, which is significantly below the US Department of Energy (DOE)'s target of US\$ 2 *per* kg of H₂[34]. The stability of the Er-RuOₓ||Pt/C PEMWE device was evaluated at 200, 500, and 1000 mA cm⁻², respectively, and no significant activity decay was observed over the device after a 100-h stability test for each condition (Fig. 3g). The degradation rate at 500 mA cm⁻² is mere 37 μV·h⁻¹, demonstrating the application potential of Er-RuOₓ for green hydrogen production.

## Origin of the excellent performance of Er-RuOₓ

X-ray photoelectron spectroscopy (XPS) was conducted to determine the chemical composition and valence states of the as-prepared catalysts. In the Ru 3*p* XPS spectrum (Supplementary Fig. 27), the peaks at 463.7 and 463.1 eV can be attributed to Ru³⁺ and Ru⁴⁺, respectively[35]. Compared with RuO₂, the Ru⁴⁺ ratio in Er-RuOₓ is lower than that in RuO₂, indicating a lower oxidation state of Ru in Er-RuOₓ. To explore the interaction of Er and Ru, X-ray absorption spectroscopy (XAS) was utilized to probe the atomic and coordination environment. The X-ray absorption near-edge structure spectra (XANES) of Ru K-edge (Fig. 4a) reveal that the absorption threshold position of Er-RuOₓ is higher than that of the Ru foil but lower than that of RuO₂. The calculated oxidation state of Ru in Er-RuOₓ is 3.80 (Supplementary Fig. 28), consistent with the results of XPS and Bader charge. Moreover, the Ru K-edge Fourier transform extended X-ray absorption fine structure (FT-EXAFS) spectrum of Er-RuOₓ (Fig. 4b) displays a peak located at ~1.98 Å, assigned to the Ru–O bonds. This peak exhibits a positive shift towards longer interatomic distances in comparison to that of RuO₂ because of the slightly weakened Ru–O covalency (Supplementary Table 9), matching well with theoretical calculation predictions. The Wavelet transform analysis also demonstrates that Ru–O coordination in Er-RuOₓ is similar to that of RuO₂, while no Ru–Ru bond is detected in Er-RuOₓ, aligning with the FT-EXAFS results (Fig. 4g–i).

We then conducted the in situ Raman spectroscopy measurement (Supplementary Fig. 29) to elucidate the structure evolution during

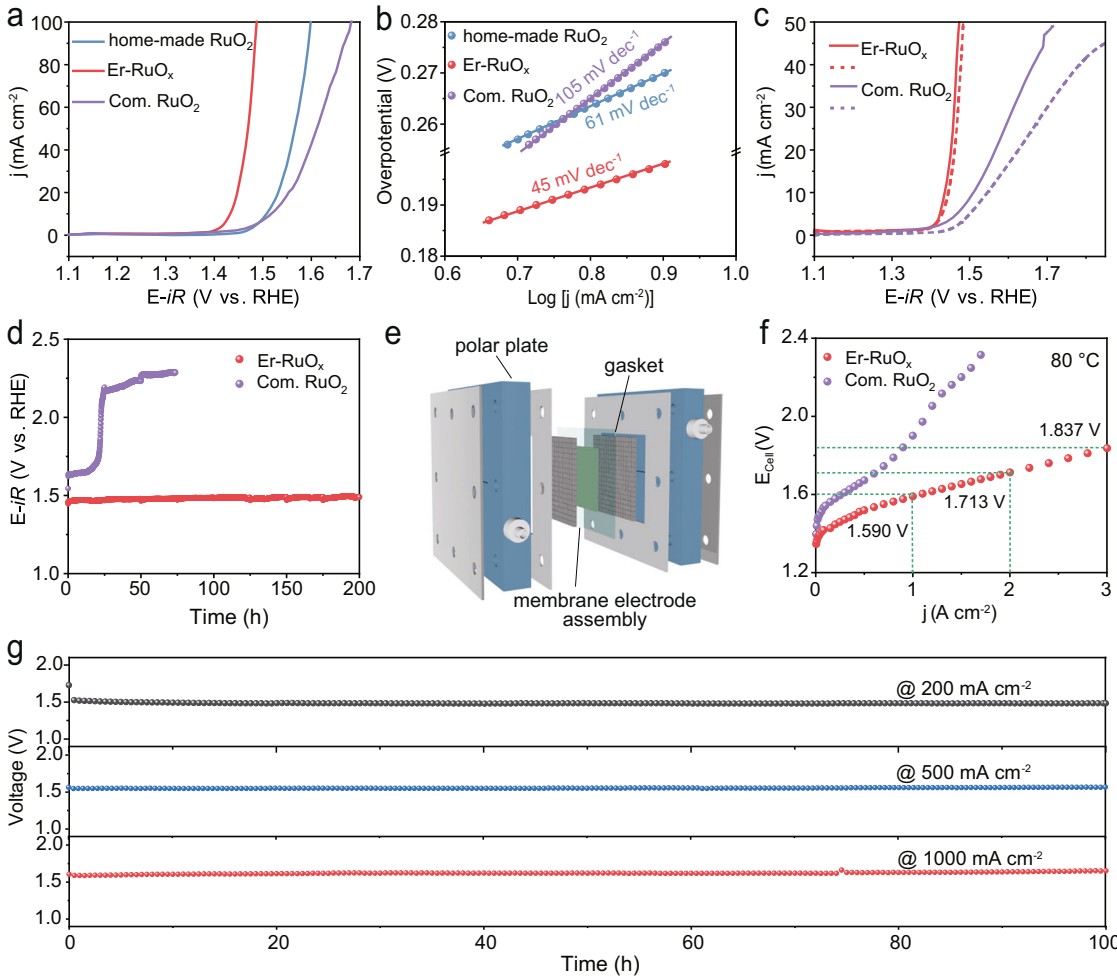

**Fig. 3 | Electrocatalytic performance of the as-prepared catalysts. a** OER polarization curves of home-made $RuO_2$, Er-$RuO_x$ and commercial $RuO_2$ with *iR*-corrected (95%), where *R* was measured to be $1.1 \pm 0.1\ \Omega$. **b** Tafel plots of home-made $RuO_2$, Er-$RuO_x$ and commercial $RuO_2$. **c** Polarization curves of Er-$RuO_x$ and commercial $RuO_2$ before (solid line) and after (dash line) 30,000 CV cycles. **d** The CP curves of Er-$RuO_x$ and commercial $RuO_2$ at 10 mA cm$^{-2}$. **e** Schematic diagram of the PEMWE electrolyzer. **f** Polarization curves of the Er-$RuO_x$ and commercial $RuO_2$-based PEMWE at 80 °C. **g** The CP curves of Er-$RuO_x$-based PEMWE electrolyzer operated at 200, 500, and 1000 mA cm$^{-2}$, respectively.

OER. Raman spectra shows the nanocrystalline nature of $RuO_2$ and Er-$RuO_x$ (Fig. 4c, d), with two major peaks located at ~430 and 600 cm$^{-1}$, corresponding to the $E_g$ and $A_{1g}$, respectively. As the potential increases from open circuit potential (OCP) to 1.8 V vs. RHE, Er-$RuO_x$ maintains a nearly constant Raman shift, indicating the stability of Ru−O configurations. In contrast, a ~10 cm$^{-1}$ red shift is observed in $RuO_2$, suggesting the shrinkage in Ru−O bonding length during OER.

Furthermore, Ru K-edge XANES of Er-$RuO_x$ and $RuO_2$ were collected with applied potential rise from 1.3 to 1.7 V vs. RHE to investigate the changes in local Ru atomic structure and chemical coordination. As presented in Fig. 4e, f and Supplementary Fig. 30, Er-$RuO_x$ and $RuO_2$ exhibit significant differences in the variation of Ru oxidation sates, especially at high bias. In detail, when the applied voltage is 1.3 V, the oxidation state of Ru in Er-$RuO_x$ increases from 3.80 to 4.16, followed by a negligible change when potential further increased to 1.7 V. In contrast, the oxidation states of Ru in $RuO_2$ steadily increases to 4.36 as the applied bias varies from 1.3 to 1.7 V, showing no sign of stabilization. The result suggests that the stable structure of Er-$RuO_x$ can be maintained at high oxidation potentials, whereas $RuO_2$ undergoes significant structural evolution. The Ru 3*d* peak in Er-$RuO_x$ XPS spectrum after stability test is slightly positive-shift (Supplementary Fig. 31), indicating that the surface of Er-$RuO_x$ is oxidized

to a higher valence state, consistent with the XANES results. TEM images after the stability test confirm that the crystalline structure of Er-$RuO_x$ (Supplementary Fig. 32) is well preserved, indicating no significant structure deterioration occurred under the strong oxidative bias.

## Adsorption behavior analysis of Er-$RuO_x$ with oxygen intermediates

To rationalize the improved OER performance on Er-$RuO_x$, the adsorptions of Er-$RuO_x$ and $RuO_2$ to oxygen intermediates were evaluated. As shown in Fig. 5a, Er-$RuO_x$ exhibits a lower d-band center of Ru (−1.945 eV) than that of the pure $RuO_2$ (−1.900 eV), indicating that the 4*f*-2*p*-4*d* orbital hybridization slightly changes the electronic environment of Ru *d* orbitals. Further, the $d_{z^2}$ orbitals can be hybridized into $\sigma$ and $\sigma^*$ with *OH intermediates. The Ru $d_{z^2}$-state energy level (−0.855 eV) of Er-$RuO_x$ up-shifts to the Fermi level relative to that of the $RuO_2$ (−2.171 eV) (Fig. 5b), thereby resulting in less electronic occupancy in the antibonding states and stronger *OH adsorption (Fig. 5c). As shown in Fig. 5d, more charge accumulation toward *OH verifies a stronger electron transfer behavior between Er−O−Ru and *OH, confirming the strengthened *OH adsorption.

The strengthened *OH adsorption was also experimentally verified using methanol as a probe. The methanol oxidation reaction

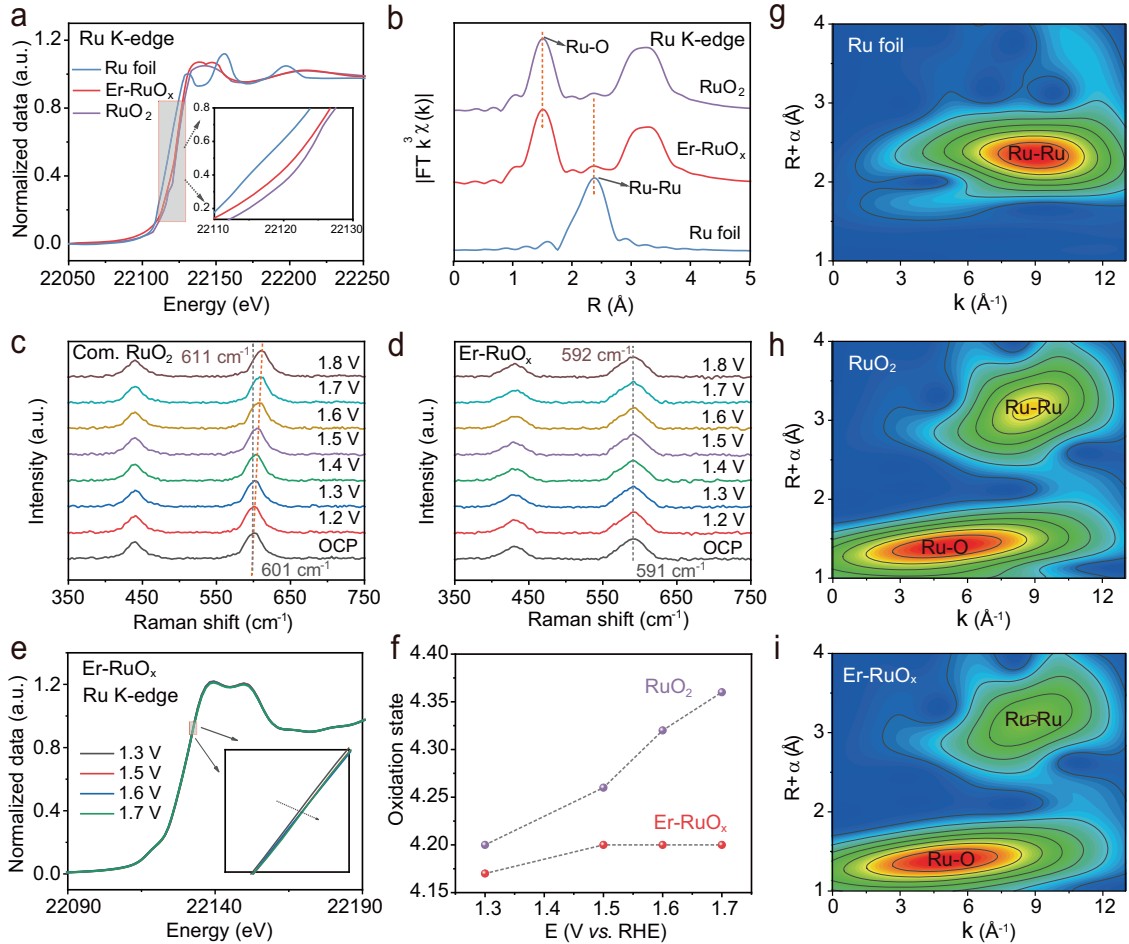

**Fig. 4 | Structural characterization of Er-RuOₓ and RuO₂. a** Ru K-edge XANES spectra of Ru foil, RuO₂, and Er-RuOₓ. **b** EXAFS spectra of Ru foil, RuO₂, and Er-RuOₓ. In situ Raman spectra obtained under various applied potential on (**c**) RuO₂ and (**d**) Er-RuOₓ. **e** Ru K-edge XANES spectra of Er-RuOₓ with different applied bias. **f** The variation trend of Ru oxidation states in RuO₂ and Er-RuOₓ under different potentials. WT-EXAFS of (**g**) Ru foil, (**h**) RuO₂, and (**i**) Er-RuOₓ.

(MOR) follows a well-established mechanism, in which methanol molecules tend to nucleophilically attack the electrophilic *OH. As a result, MOR is more active on surfaces with stronger *OH adsorption[36]. When 1.0 M methanol was introduced into the 0.5 M $H_2SO_4$ solution, the current densities of Er-RuOₓ and RuO₂ showed a substantial increase compared to those before the addition of methanol, attributed to the methanol electrooxidation (Fig. 5e, f). The difference in current densities induced by MOR, which was directly proportional to the number of charges transferred, was quantified by calculating the filled area between the curves. The bigger current difference observed between the MOR and OER over Er-RuOₓ than that of RuO₂ suggested its stronger MOR competition reaction, verifying the enhanced *OH adsorption on Er-RuOₓ (Fig. 5g).

To acquire the information of oxygen intermediates for a more comprehensive mechanistic understanding, the in situ attenuated total reflection-surface enhanced infrared absorption spectroscopy (ATR-SEIRAS) was performed using a home-made electrochemical cell. The ATR-SEIRAS of both RuO₂ and Er-RuOₓ were measured between 1000 and 1400 cm⁻¹ from 1.3 V to 1.7 V vs. RHE. As shown in Fig. 5h, a potential-dependent peak at 1211 cm⁻¹ for RuO₂, corresponding to the stretching vibration of *OOH species[18], became more prominent as the potential increased from 1.5 V to 1.7 V. For Er-RuOₓ, the peak of *OOH species was emerged at 1.4 V (Fig. 5i), in accordance with the lower PDS of *O to *OOH (Fig. 1d). Meanwhile, the emergence of *OOH species indicated that the adsorbate evolution mechanism pathway dominated O₂ generation over Er-RuOₓ.

## Discussion

In summary, we demonstrate that precisely tuning of the Ru−O covalency in RuOₓ can be achieved by introducing Ln elements through *d-p-f* orbital hybridization. Benefiting from the optimized Ru−O covalency, Er-RuOₓ is screened out and exhibits the high catalytic stability, significantly outperforming pristine RuO₂ by orders of magnitude. The operando characterizations indicate that the oxidation state of Ru in Er-RuOₓ initially increases as the applied potential increases and then remains nearly constant. In contrast, the oxidation state of Ru in RuO₂ continuously rises with no sign of stabilization. DFT calculations and methanol molecular probe experiments validate the stronger *OH adsorption on Er-RuOₓ relative to that on RuO₂, thus leading to the enhanced OER activity. The PEMWE (Nafion 117 membrane) employing Er-RuOₓ as the anode catalyst requires only 1.837 V to reach 3 A cm⁻² and exhibit long-term stability at 500 mA cm⁻² for 100 h with a degradation rate of mere 37 μV h⁻¹. This study provides a principled catalyst design framework for the precise Ru−O covalency control, thereby guiding the development of ruthenium-based catalysts suitable for practical implementation in PEMWE systems.

## Methods
### Materials
Ruthenium chloride (RuCl₃, 99.5%) was purchased from Beijing Inno-Chem Science & Technology Co., Ltd. Holmium nitrate pentahydrate (Ho(NO₃)₃·5H₂O, 99.99%), erbium nitrate pentahydrate (Er(NO₃)₃·5H₂O, 99.9%) and thulium nitrate hexahydrate (Tm(NO₃)₃·6H₂O, 99.9%) was

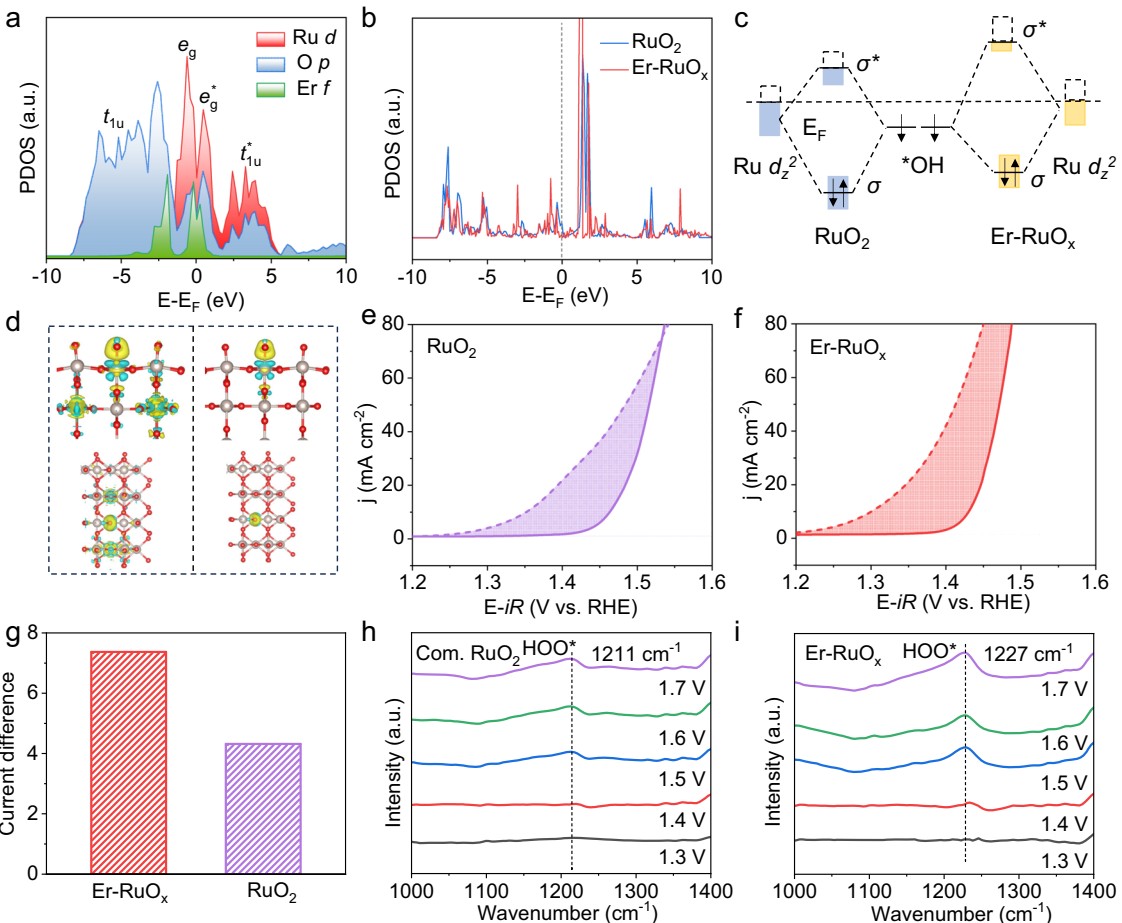

**Fig. 5 | Analysis of adsorption behavior. a** The calculated PDOS of Ru $d$, O $p$, and Er f orbitals of Er-RuO$_x$. **b** The PDOS of Ru $d_{z^2}$ orbital of RuO$_2$ and Er-RuO$_x$. **c** Schematic diagram of orbital hybridization between $d_{z^2}$ orbital of RuO$_2$ and Er-RuO$_x$ and the *OH bonding orbitals. **d** Charge density difference of *OH-adsorbed Er-RuO$_x$ (left) and RuO$_2$ (right). Yellow and blue iso-surfaces represent charge accumulation and depletion, respectively. Polarization curves of (**e**) RuO$_2$ and (**f**) Er-RuO$_x$ in 0.5 M H$_2$SO$_4$ solution with (dashed lines) and without (solid lines) 1 M methanol. **g** Current difference between the polarization curves in 0.5 M H$_2$SO$_4$ solution with and without 1.0 M methanol for Er-RuO$_x$ and RuO$_2$. ATR-SEIRAS analysis of (**h**) commercial RuO$_2$ and (**i**) Er-RuO$_x$.

purchased from Shanghai Macklin Biochemical Co., Ltd. Urea (AR) and glucose (AR) were obtained from Beijing Tong Guang Fine Chemicals Company. Commercial Pt/C (70% Pt) was obtained from Johnson Matthey Company. Nafion 117 membrane (thickness: 0.18 mm) was obtained from DuPont Co. All reagents were used without further purification.

### Synthesis of Ln-RuO$_x$ and RuO$_2$

In the typical synthesis of Ln-RuO$_x$, 40 mg RuCl$_3$ and a certain amount of Ln(NO$_3$)$_3$ were added into 5 mL deionized water with 1 g urea and 5 g glucose (keeping the atomic ratio of Ln:Ru at 1:10). The mixture was stirred until a homogeneous solution was attained. Subsequently, it was subjected to heating at 140 °C for 8 h in an oven to form a porous foam, and then annealed at 500 °C for 10 h in the air to obtain Ln-RuO$_x$. For comparison, pure RuO$_2$ was also prepared without the addition of Ln(NO$_3$)$_3$.

### Electrochemical measurements

All electrochemical measurements were conducted with the CHI 760E electrochemical workstation employing a three-electrode cell at room temperature. The as-prepared catalysts (2 mg) were dispersed in a mixture of 980 µL isopropanol and 20 µL Nafion D-521 (5 wt%) solution. After ultrasonication for 1 h, the homogeneous ink was carefully dropped onto the carbon paper (1 cm$^2$) to obtain the working electrode with a desirable loading of 0.5 mg cm$^{-2}$. The

reference electrode was Hg/Hg$_2$SO$_4$, calibrated in a three-electrode system in which Pt wires served as both working electrode and counter electrode, and H$_2$-saturated 0.5 M H$_2$SO$_4$ solution was employed as the electrolyte. Subsequently, CV was measured at a scan rate of 1 mV s$^{-1}$. The average potential at which the current crosses zero was determined as the thermodynamic potential relative to Hg/Hg$_2$SO$_4$[33]. All potentials were calibrated to RHE by the equation: E (vs. RHE) = E (vs. Hg/Hg$_2$SO$_4$) + 0.707 V. The graphite rod was used as the counter electrode. The polarization curves were performed at a scan rate of 5 mV s$^{-1}$ in 0.5 M H$_2$SO$_4$ solution (pH ≈ 0.3). All linear sweep voltammetry (LSV) curves measured in three-electrode cell were iR-corrected (95%) unless otherwise stated, where R was measured to be 1.1 ± 0.1 Ω. During electrolyte preparation, 13.6 mL sulfuric acid (98%) was added to a beaker containing a suitable volume of deionized water, followed by adjustment of the volume to 500 mL in a volumetric flask and shake to ensure thorough mixing. The electrolyte is freshly prepared and promptly utilized. Electrochemical impedance spectroscopy was performed at 1.485 V vs. RHE in the frequency range from 10$^{-2}$ Hz to 10$^5$ Hz. CVs at various scan rates (namely, 10, 20, 30, 40, and 50 mV s$^{-1}$) were performed to calculate the electrochemical active surface area (ECSA), which was proportional to the double layer capacitance (C$_{dl}$). Assuming that the specific capacitance of a flat surface was ~40 µF for 1 cm$^2$ of real surface area, then the ECSA was estimated as: $ECSA = \frac{C_{dl}(mF cm^{-2})}{40\mu F cm^{-2} per cm^2_{ECSA}}$.

## PEMWE tests

The PEMWE was assembled with Er-RuO$_x$ or commercial RuO$_2$ as anode, with a loading of ~3 mg cm$^{-2}$. Commercial Pt/C (70 wt% Pt) was utilized as the cathode catalyst (~0.5 mg$_{Pt}$ cm$^{-2}$). Pt-coated Ti fiber was used as gas diffusion layers. Nafion 117 was used as the proton exchange membrane (PEM), which was sequentially treated with H$_2$O$_2$ and 0.5 M H$_2$SO$_4$ at 80 °C for 1 h. The anode and cathode catalyst ink were separately prepared by dispersing the catalyst in a mixture of Nafion (5%), isopropanol, and distilled water. All the cathode ink was sprayed onto the PEM. As for anode, half of catalyst ink was sprayed onto the PEM, while the remaining half was sprayed onto the surface of gas diffusion layer. The catalyst-loaded PEM and gas diffusion layers were then hot-pressed together at 120 °C for 2 min under a pressure of 2 MPa to fabricate the membrane electrode assembly (MEA), which was sandwiched by two Ti bipolar plates to complete a PEMWE device. Each Ti bipolar plate featured a serpentine flow channel with 1 cm × 1 cm reactive area. The PEMWEs were operated at 80 °C utilizing distilled water as the electrolyte, which was delivered to the anode by a peristaltic pump. All measurements in PEMWEs were recorded without iR-correction.

## In situ ATR-SEIRAS measurements

The in situ ATR-SEIRAS measurement was carried out on Bruker 70 V Fourier-transform infrared (FTIR) spectrometer. The measurement featured a spectral resolution of 8 cm$^{-1}$, with 64 interferograms co-added for each spectrum. The preparation of working electrodes comprised of two steps: firstly, an ultra-thin Au film was chemical deposited on the Si crystal to enhance the IR signal and facilitate electron conduction; secondly, the catalyst ink was dropped onto the Au film with a loading of 0.1 mg cm$^{-2}$. The Si crystal loaded with catalyst was placed onto a spectro-electrochemical three-electrode cell. Ag/AgCl electrode and Pt wire served as the reference and counter electrodes, respectively. The 0.1 M HClO$_4$ solution was used as the electrolyte. All SEIRAS spectra were obtained during the LSV test.

## DFT calculations

The calculations were performed employing the density functional theory as implemented in the Vienna ab initio simulation package (VASP)[37]. The exchange-correlation function was described using the generalized gradient approximation (GGA)[38] parameterized by the Perdew–Burke–Ernzerhof (PBE). The cut-off energy for the plane wave basis was set to 450 eV. RuO$_2$(110) surface (Supplementary Fig. 1) was modeled by a 2 × 2 × 2 supercell (with 55 O atoms and 24 Ru atoms), in which the top three atomic layers are allowed to relax. As for Ln-RuO$_x$, two Ru atoms in the RuO$_2$(110) model was substituted by Ln atoms (Supplementary Figs. 2–4 and Supplementary Data 1), leading to a chemical composition of 8.3 at percentage Ln and 91.7 at percentage Ru. A vacuum spacing of 20 Å was set along the z-direction to prevent the interaction between the slab and its periodic motif. The termination of RuO$_2$(110) and Ln-RuO$_x$ has all Ru filled with oxygen but one coordinatively unsaturated site Ru unfilled, which is the active site for intermediates adsorption. The Monkhorst-Pack method was used for sampling the Brillouin zone with a 3 × 3 × 1 mesh. The geometry relaxation and convergence criteria for the electronic structure were 0.05 eV/Å and 5 × 10$^{-5}$ eV, respectively. The free energies of the reaction steps were calculated by the equation: $\Delta G_{ads} = \Delta E_{ads} + \Delta E_{ZPE} - T\Delta S$, where $\Delta E_{ads}$ is the adsorption energy of intermediates, and $T$ is temperature. $\Delta E_{ZPE}$ and $\Delta S$ represent the energy variance in zero-point energy and entropy, respectively. When calculating the formation energy of Ru vacancies, the lost Ru atoms originate from unsaturated surface coordination sites, consistently from the same position across all models. In the calculation of oxygen vacancy formation energy, both Ln-RuO$_x$ and RuO$_2$(110) lose oxygen atoms surrounding unsaturated Ru coordination sites on the surface. ICOHP and Bader charge represent the values at active site.

## Data availability

The data that support the conclusions of this study are available within the paper and Supplementary information. Source data are provided with this paper.

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

## Acknowledgements

S.J.G. acknowledge the fundings from National Natural Science Foundation of China (Nos. 52333010, 52025133, 22309004, 52261135633, 52303363, 52302207, 22205010, 22305010, 22105007), National Key R&D Program of China (No. 2022YFE0128500), China National Petroleum Corporation-Peking University Strategic Cooperation Project of Fundamental Research, the Beijing Natural Science Foundation (No. Z220020), New Cornerstone Science Foundation through the XPLORER PRIZE, CNPC Innovation Found (No. 2021DQ02-1002), China National Postdoctoral Program for Innovative Talents (No. BX20220009), China Postdoctoral Science Foundation (Nos. 2022M720225, 2023M730029, 2022M710187, 2023M730051, 2020M670018) and Yunnan Fundamental Research Projects (grant NO. 202401AT070370). This work was carried out with the support of 1W1B beamline at Beijing Synchrotron Radiation Facility. The authors thank the photoemission photoendstations BL14W1 in the Shanghai Synchrotron Radiation Facility (SSRF) for the help with characterizations.

## Author contributions

S.J.G. conceived and supervised the project. M.C.L. guided and supervised the whole research. L.L. and G.W.Z. performed the experiments, collected and analyzed the data. C.H.Z., F.L., H.L. and D.W.W. participated in the PEMWE tests. Y.J.T., Y.H., Y.X.L. and C.S.S. helped with the Operando ATR-SEIRAS measurements. L.Y.Z., Q.Z.H., R.J.Z. and N.Y. participated in part of basic experiments. L.L. wrote the manuscript. All authors took part in the discussion of data and gave comments on the manuscript.

## Competing interests

The authors declare no competing interests.
