## [Peer Review File · Nature Communications]

Lanthanide-regulating Ru-O covalency optimizes acidic oxygen evolution electrocatalysisEditorial Note: Parts of this Peer Review File have been redacted as indicated to remove third-party material where no permission to publish could be obtained.

REVIEWER COMMENTS

Reviewer #1 (Remarks to the Author):

The manuscript demonstrates that lanthanide elements with 4f orbital buried by 5s/5p can minimize external influences and enable precise tuning of Ru-O covalency for durable OER. The Er-RuO_x-based PEMWE requires only 1.837 V to reach 3 A cm⁻², as well as long-term stability at large current density (200, 500 and 1000 mA cm⁻²). The work is original and deserves to be published. This work might be significantly improved if the author considers the following points:

1. Page 2: "Weakening the Ru-O bond covalency can localize O 2p and Ru 3d orbitals below Fermi level, inhibiting lattice oxygen's participation in OER and the formation of oxygen vacancies, thereby preventing excessive overoxidation of Ru species into soluble RuO₄ during OER": the statement may not be entirely convincing, as Shao-Horn and Koper have reported that evolved O₂ from RuO₂ does not originate from oxygen within the lattice.
2. The authors should clarify how they can sustain OER below the thermoneutral potential, as this contradicts thermodynamic principles.
3. The computational methods should include more details, and it is recommended to provide ΔE_{ZPE} and ΔS for various models.
4. Page 5: "As Ln dissolution would take place during OER process, defective structures containing Ln vacancies were also constructed". The authors are suggested to provide evidence to support the statement.
5. It is suggested to include the following references in the manuscript, such as Nat. Synth., 2024, 10.1038/s44160-023-00453-w.

Reviewer #2 (Remarks to the Author):

The authors present an interesting work on establishing an effective strategy for stabilizing Ru-based OER electrocatalysts, instead of the conventional trial-and-error approach. Taking advantage of the shielding effect of Ln elements, the authors realized the precise regulation of Ru-O covalency. The optimal Er-RuO_x catalyst exhibits high stability for OER, which is 35.5 times higher than that of RuO₂. This work would be of broad interest to the research community. Thus I recommend the acceptance of this study after the authors address the following points:

1. In abstract part, the authors claimed that "The estimated hydrogen production cost (US\$ 0.85 per kg H₂) at 1 A cm⁻² over Er-RuO_x-based PEMWE...". How did the authors calculate the cost?
2. In Page 10, Line 10, "the calculated oxidation of Ru in Er-RuO_x is 3.80". The author should provide a detailed method for obtaining the oxidation states.
3. The authors investigated the strength of *OH adsorption by employing methanol as a probe. The reason for using methanol as a molecular probe to assess OH adsorption strength should be elucidated. The authors are urged to provide a comprehensive explanation of the methodology and mechanisms

involved in this approach.

4. Why did the author select only a few of Ln elements to explore the effects on Ru-O covalency? What is the selection principle?

Reviewer #3 (Remarks to the Author):

In this manuscript, the authors present a fine-tuning strategy involving Er doping in RuO₂ to enhance its activity and stability in the acidic OER. Using DFT calculations, the authors perform a comparative analysis of three Ln-elements (Tm, Er, and Ho) to evaluate their effects on Ru-O covalency, vacancy formation energies, and free energy profiles of OER. Among these elements, Er emerges as the most promising, leading to a subsequent experimental synthesis. A comprehensive series of characterizations confirms the superior performance of Er-doped RuO₂ compared to pure RuO₂. However, despite the promising performance in acidic OER, I have some concerns about the completeness of the control experiments and inaccuracies in the mechanistic explanations.

1. In the introduction section, the explanation of the motivation for this work seems somewhat understated. While the overall goal seems to be the identification of an effective catalyst for acidic OER, the current statements in the introduction give the impression that the primary goal is merely the continuous tuning of the Ru-O bond. While it's plausible that identifying a trend or descriptor could contribute to subsequent material design, the pursuit of continuous tuning should be positioned as a methodological or intermediate step rather than the ultimate goal. Advancing one step beyond the current motivation would enhance the clarity and purpose of the research.

2. The results in Fig. 1b-c are not consistent with the description in the manuscript. In Fig.1b-c, the -ICOHP of Ho is larger than that of Tm, but in the main text, the -ICOHP of Ho is smaller (1.619 vs. 1.622). Unfortunately, the volcano trend would no longer exist.

3. Why choose Tm/Er/Ho? The changes between these three elements are so small that they raise concerns about error variations in the results, such as 1.52/1.50/1.48 (Bader charge), -1.622/-1.636/-1.619 (ICOHP), 0.54/0.58/0.60 (Δ GO vacancy), and 2.49/2.48/2.57 (Δ GRu vacancy).

4. Furthermore, in comparison of Bader charges (Tm>Er>Ho) and -ICOHP (Er>Ho>Tm), the trends are not the same (whether in ascending or descending order), which further makes the small fluctuations appear to be due to calculation error rather than a fine-tuning result. In principle, the ICOHP is mainly influenced by the degrees of electron occupancy when considering a set of similar models, which would be expected to match the amount of electrons transferred, but the following manuscript concentrates only on the ICOHP results, where possible reasons for the different trends are not discussed.

5. The trend of Ru vacancy formation energies is reversed before and after Ln leaching, with Δ GRu vacancy (Er<Ho<Tm) and Δ GRu vacancy after Ln-leaching (Er>Ho>Tm), so what about Δ GO vacancy?

Would it also be reversed? If so, does it mean that the stability of O and Ru cannot be achieved simultaneously? At present, the authors chose ΔG_{O} vacancy before Ln leaching (Fig. 1b) and ΔG_{Ru} vacancy after Ln leaching (Fig. 1c) to conclude, but the formation energies of these two kinds of vacancies should be compared under the same model in principle.

6. Please specify the atom sites which are considered in these calculations. Are these results from a single site or averaged over a number of sites? Are the atoms in the surface layer, sublayer, or full slab?

7. Care should be taken when using expressions such as “the most stable catalyst”. From the Supplementary Tables 1 and 2, it seems that there is no significant difference in the performances between Er-RuO_x and Ni-RuO_x (Nat. Mater. 2023,22, 100-108), except for a 14mV decrease in η for the latter. Given that Ni is much more abundant than Er, what are the advantages of using Er?

8. Considering the possible different synthesis/characterization procedures, it might be more convincing to include the control experiments of d-element doping, such as Ni. Also, if one wants to emphasize continuous tuning when using f-elements to dope RuO_x, it would be better to include control experiments with different f-element doped samples.

9. To obtain information about the occupancy of bonding/antibonding states, the results of COHP between *OH and the adsorption site (Ru) could provide more direct evidence compared to the dz² band center.

Reviewer #4 (Remarks to the Author):

The authors used lanthanide-group (Ln) elements to regulate ruthenium (Ru)-oxygen (O) covalency to enhance the electrocatalytic stability of RuO_x. The authors claimed the 4d-2p-4f orbital hybridization resulted in the improved both oxygen evolution reaction (OER) activity and stability. This work is interesting and may attract community attention. Nevertheless, some revisions are required prior to further consider in Nature Communications.

1) Since the formation energy trends of RuO_x doped with Ho, Tm, and Er in Figure 1b and c are different, the additional discussion is necessary for the variation.

2) It is necessary to explain why only Ho, Er, and Tm were chosen for the calculation and comparison among various Lanthanide-group elements.

3) While Er-RuO_x theoretically expected to exhibit the best activity and stability, and it was experimentally validated. However, Ho-RuO_x and Tm-RuO_x should also be additionally synthesized to experimentally demonstrate that their OER activity and stability are superior to RuO₂ but inferior to Er-RuO_x.

4) It would be beneficial to include control experiments to determine the optimum Er ratio.

5) The claim of successfully achieving hydrogen generation at a lower cost according to the US

Department of Energy (DOE) standards lacks explanation in this paper. Please add relevant data comparison.

6) It is recommended to organize the figures. Upon reviewing Supporting Figures 9 and 10, the colors of the samples in the supporting are different from the main text.

7) Check and correct references and typos.

A point-by-point response to the reviewers' comments

To Reviewer 1:

Overall comments: The manuscript demonstrates that lanthanide elements with 4f orbital buried by 5s/5p can minimize external influences and enable precise tuning of Ru-O covalency for durable OER. The Er-RuO_x-based PEMWE requires only 1.837 V to reach 3 A cm⁻², as well as long-term stability at large current density (200, 500 and 1000 mA cm⁻²). The work is original and deserves to be published. This work might be significantly improved if the author considers the following points:

Response: We appreciate your kind support for our work. According to your valuable comments, we have revised our manuscript (marked in red in the revised version) and responded point-by-point as follows.

Comment 1: Page 2: “Weakening the Ru-O bond covalency can localize O 2p and Ru 3d orbitals below Fermi level, inhibiting lattice oxygen’s participation in OER and the formation of oxygen vacancies, thereby preventing excessive overoxidation of Ru species into soluble RuO₄ during OER”: the statement may not be entirely convincing, as Shao-Horn and Koper have reported that evolved O₂ from RuO₂ does not originate from oxygen within the lattice.

Response 1: Thank you for your valuable comment. As Shao-Horn and coworkers reported in ACS Energy Letters 2017, there was no oxygen exchange on the ideal RuO₂(100), (110), (101), and (111) film surfaces in both basic and acidic environments. However, it was experimentally observed for nanocrystalline RuO₂-based catalysts that the involvement of lattice oxygen in the OER becomes pronounced at potentials above 1.12 V (Electrochem. Commun. 2009, 11, 1865-1868). The discrepancy between the two sets of experimental results was due to different chemistries of the surface-active sites and surface crystallinity of the as-prepared catalysts (ACS Catal. 2020, 10, 3650-3657). As demonstrated by Alexandrov and coworkers, structural defects can make the lattice oxygen mechanism (LOM) be competitive with the conventional adsorbate evolving mechanism (AEM) of the OER in rutile RuO₂ catalysts (ACS Catal. 2020, 10, 3650-3657). This interprets why the involvement of lattice oxygen was experimentally observed for the nanocrystalline RuO₂-based materials, but not for the ideal metal-oxide films. In order to be more rigorous, we have revised the corresponding description in the manuscript.

Revision: Line 52-55. “The stability of nanocrystalline RuO₂-based catalysts is closely tied to the covalency of Ru-O bonds⁶. Weakening the Ru-O bond covalency can localize O 2p and Ru 3d orbitals below Fermi level, inhibiting lattice oxygen’s participation in OER and the formation of oxygen vacancies, thereby preventing excessive overoxidation of Ru species into soluble RuO₄ during OER²⁴.”

Comment 2: The authors should clarify how they can sustain OER below the thermoneutral potential, as this contradicts thermodynamic principles.

Response 2: Thank you for your valuable comment.

The equilibrium potential U_{rev}^0 to perform the electrolysis of water is given by:

$$U_{rev}^0 = \frac{\Delta G^0}{n \cdot F} = \frac{237000 \text{ J/mol}}{2 \cdot 96485 \text{ C/mol}} = 1.23 \text{ V} \quad (1)$$

Where n is the number of charges transferred *per* hydrogen molecule ($n = 2$) and F is Faraday's constant (96485 C mol^{-1}), $\Delta G^0 = 237 \text{ kJ mol}^{-1}$. The thermoneutral potential U_{th}^0 is connected to the overall change in enthalpy ΔH through the equation 2, $\Delta H^0 = 286 \text{ kJ mol}^{-1}$ (International Journal of Hydrogen Energy, 2014, 39, 9457-9466)

$$U_{th}^0 = \frac{\Delta H^0}{n \cdot F} = \frac{286000 \text{ J/mol}}{2 \cdot 96485 \text{ C/mol}} = 1.48 \text{ V} \quad (2)$$

[REDACTED]

Figure R1. Cell potential for hydrogen production by water electrolysis as a function of temperature (Progress in Energy and Combustion Science, 2010, 36, 307-326).

Table R1. Table of electrolysis in adiabatic conditions.

State	Cell voltage (E)	electrolyzer	Energy transfer	environment	Can electrolysis proceed?
1	<1.23 V	electrolyzer temperature	No	room temperature	No
2	1.23 V < E < 1.48 V				No
3	≥1.48 V				Possible

Figure R1 shows the relationship between the electrolyzer cell potential and operating temperature. The cell potential-temperature plane is divided into three zones by the equilibrium voltage line and thermoneutral voltage line. The equilibrium voltage is the theoretical minimum potential required to dissociate water by electrolysis, below which the electrolysis of water cannot proceed. Above the equilibrium potential line, the electrolysis is possible to proceed. it is divided into endothermic and exothermic reactions by the thermoneutral voltage line. When the electrolysis process is performed under adiabatic conditions (Table R1), the total reaction enthalpy ($T\Delta S \sim 49 \text{ kJ mol}^{-1}$) must be provided by the electrical current. Under this circumstance, the thermoneutral voltage (1.48 V) is required to maintain the electrochemical reaction without heat generation or adsorption. Since the actual measurement environment is not adiabatic, the electrolyzer will exchange energy with the environment during the test, so the required voltage does not have to be higher than 1.48 V (Table R2).

Table R2. Energy transfer of electrolysis.

State	Cell voltage (E)	electrolyzer	Energy transfer	environment	Can electrolysis proceed?
1	<1.23 V	room temperature		room temperature	No
2	1.23 V < E < 1.48 V	<room temperature, adsorption energy from environment			Possible
3	>1.48 V	>room temperature, release energy to environment			Possible

Comment 3: The computational methods should include more details, and it is recommended to provide ΔE_{ZPE} and T ΔS for various models.

Response 3: Thank you for your valuable suggestion. We have provided computational methods details in the revised manuscript. Additionally, we provided ΔE_{ZPE} and T ΔS of different models in the Supplementary Information.

Revision: Line 349-357. “RuO₂(110) surface (Supplementary Fig. 1) was modeled by a 2×2×2 supercell (with 55 O atoms and 24 Ru atoms), in which the top three atomic layers are allowed to relax. As for Ln-RuO_x, two Ru atoms in the RuO₂(110) model was substituted by Ln atoms (Supplementary Figs. 2-4), leading to a chemical composition of 8.3 at percentage Ln and 91.7 at percentage Ru. A vacuum spacing of 20 Å was set along the z-direction to prevent the interaction between the slab and its periodic motif. The termination of RuO₂(110) and Ln-RuO_x has all Ru filled with oxygen but one coordinatively unsaturated site Ru unfilled, which is the active site for intermediates adsorption. The Monkhorst-Pack method was used for sampling the Brillouin zone with a 3×3×1 mesh.”

Line 361-365. “When calculating the formation energy of Ru vacancies, the lost Ru atoms originate from unsaturated surface coordination sites, consistently from the same position across all models. In the calculation of oxygen vacancy formation energy, both Ln-RuO_x and RuO₂(110) lose oxygen atoms surrounding unsaturated Ru coordination sites on the surface. ICOHP and Bader charge represent the values at active site.”

Supplementary Table 1 | ΔE_{ZPE} and T ΔS for each OER intermediate on RuO₂.

Models	ΔE_{ZPE} -T ΔS (eV)
RuO ₂ -*OH	0.3859

RuO ₂ -*O	0.0935
RuO ₂ -*OOH	0.4043

Supplementary Table 2 | ΔE_{ZPE} and T ΔS for each OER intermediate on Ho-RuO_x.

Models	ΔE_{ZPE} -T ΔS (eV)
Ho-RuO _x -*OH	0.4057
Ho-RuO _x -*O	0.1182
Ho-RuO _x -*OOH	0.355

Supplementary Table 3 | ΔE_{ZPE} and T ΔS for each OER intermediate on Er-RuO_x.

Models	ΔE_{ZPE} -T ΔS (eV)
Er-RuO _x -*OH	0.3743
Er-RuO _x -*O	0.1367
Er-RuO _x -*OOH	0.3611

Supplementary Table 4 | ΔE_{ZPE} and T ΔS for each OER intermediate on Tm-RuO_x.

Models	ΔE_{ZPE} -T ΔS (eV)
Tm-RuO _x -*OH	0.3912
Tm-RuO _x -*O	0.1224
Tm-RuO _x -*OOH	0.4031

Comment 4: Page 5: “As Ln dissolution would take place during OER process, defective structures containing Ln vacancies were also constructed”. The authors are suggested to provide evidence to support the statement.

Response 4: Thank you for your valuable suggestion. We measured the concentrations of dissolved Ln in Ln-RuO_x after 20-cycles accelerated durability test by ICP-MS, demonstrating preferential dissolution of Ln.

Supplementary Table 5 | The concentrations of dissolved Ln and Ru in Ln-RuO_x after 20-cycles accelerated durability test by ICP-MS.

Samples	Concentration (ug/L)
Ho-RuO _x	9.0877
Er-RuO _x	8.1903

Tm-RuO _x	10.1503
---------

Comment 5: It is suggested to include the following references in the manuscript, such as *Nat. Synth.*, 2024, 10.1038/s44160-023-00453-w.

Response 5: Thank you for your valuable suggestion. We have added the reference to the revised manuscript.

Revision: Line 378-379.

“4 Li, F. & Baek, J.-B. Active site engineering accelerates water electrolysis. *Nat. Synth.* (2024). <https://doi.org/10.1038/s44160-023-00453-w>.”

To Reviewer 2:

Overall comments: The authors present an interesting work on establishing an effective strategy for stabilizing Ru-based OER electrocatalysts, instead of the conventional trial-and-error approach. Taking advantage of the shielding effect of Ln elements, the authors realized the precise regulation of Ru-O covalency. The optimal Er-RuO_x catalyst exhibits high stability for OER, which is 35.5 times higher than that of RuO₂. This work would be of broad interest to the research community. Thus I recommend the acceptance of this study after the authors address the following points:

Response: Thank you so much for the recognition and affirmation of our work. We sincerely appreciate your professional advices and valuable comments.

Comment 1: In abstract part, the authors claimed that “The estimated hydrogen production cost (US\$ 0.85 per kg H₂) at 1 A cm⁻² over Er-RuO_x-based PEMWE....”. How did the authors calculate the cost?

Response 1: Thank you for your valuable comment. The calculation of the hydrogen production cost was provided in the Supplementary Information.

Revision: Supplementary Information

“Cost of per kilogram H₂^[1]:

Cost (H₂/kg) = energy consumption × electricity bill

$$= 42.61 \text{ kW h/kg H}_2 \times \$ 0.02/\text{kW h}$$

$$= \$ 0.85/\text{kg H}_2$$

1. Hao, S. et al. Torsion strained iridium oxide for efficient acidic water oxidation in proton exchange membrane electrolyzers. *Nat. Nanotechnol.* **16**, 1371-1377 (2021).”

Comment 2: In Page 10, Line 10, “the calculated oxidation of Ru in Er-RuO_x is 3.80”. The author should provide a detailed method for obtaining the oxidation states.

Response 2: Thank you for your valuable comment. We have added the detailed methods for obtaining the oxidation states in the Supplementary Information.

Revision: Supplementary Information.

“Calculation of oxidation states from XANES

Firstly, background deduction and normalization of $\chi\mu(E)$ data were executed before E₀ calibration with foil samples as standard references. Then E₀ of each studied sample were determined by their first derivative vertex (second peak for Ru element). Taking Ru foil and RuO₂ as references, the oxidation states of the samples with different E₀ values can be obtained through linear fitting.”

Comment 3: The authors investigated the strength of *OH adsorption by employing methanol as a probe. The reason for using methanol as a molecular probe to assess OH adsorption strength should be

elucidated. The authors are urged to provide a comprehensive explanation of the methodology and mechanisms involved in this approach.

Response 3: Thank you for your valuable comment. We have provided the detailed explanation for utilizing methanol as a molecular probe to evaluate the strength of OH adsorption in the revised manuscript.

Revision: Line 274-276. “The methanol oxidation reaction (MOR) follows a well-established mechanism in which methanol molecules tend to nucleophilically attack the electrophilic *OH. As a result, MOR is more active on surfaces with stronger *OH adsorption³⁶.”

Line 278-282. “The difference in current densities induced by MOR, which was directly proportional to the number of charges transferred, was quantified by calculating the filled area between the curves. The bigger current difference observed between the MOR and OER of Er-RuO_x than that of RuO₂ suggested its stronger MOR competition reaction, verifying the enhanced *OH adsorption on Er-RuO_x (Fig. 5g).”

Comment 4: Why did the author select only a few of Ln elements to explore the effects on Ru-O covalency? What is the selection principle?

Response 4: Thank you for your valuable comment. Lanthanide-group elements exhibit biperiodic chemical trends, with Gd serving as the dividing line, where La to Gd form one peak and Gd to Lu form another peak. This pronounced division into two groups arises from the difference in the 4f electron occupation. In order to achieve precise and monotonic modulation of Ru-O covalency, we selected the elements exhibiting linear variation in their chemical properties for the investigation.

To Reviewer 3:

Overall comments: In this manuscript, the authors present a fine-tuning strategy involving Er doping in RuO₂ to enhance its activity and stability in the acidic OER. Using DFT calculations, the authors perform a comparative analysis of three Ln-elements (Tm, Er, and Ho) to evaluate their effects on Ru-O covalency, vacancy formation energies, and free energy profiles of OER. Among these elements, Er emerges as the most promising, leading to a subsequent experimental synthesis. A comprehensive series of characterizations confirms the superior performance of Er-doped RuO₂ compared to pure RuO₂. However, despite the promising performance in acidic OER, I have some concerns about the completeness of the control experiments and inaccuracies in the mechanistic explanations.

Response: We appreciate your constructive comments, which have helped us to substantially improve the quality of manuscript. We have revised the manuscript accordingly.

Comment 1: In the introduction section, the explanation of the motivation for this work seems somewhat understated. While the overall goal seems to be the identification of an effective catalyst for acidic OER, the current statements in the introduction give the impression that the primary goal is merely the continuous tuning of the Ru-O bond. While it's plausible that identifying a trend or descriptor could contribute to subsequent material design, the pursuit of continuous tuning should be positioned as a methodological or intermediate step rather than the ultimate goal. Advancing one step beyond the current motivation would enhance the clarity and purpose of the research.

Response 1: Thanks very much for the valuable suggestion. According to the suggestion, we have revised the introduction, emphasizing that the continuous regulation of Ru-O covalency aims to achieve a more promising RuO₂-based catalyst.

Revision: Line 64-66. "Herein, we reason that lanthanide (Ln)-group elements with the 4f orbital buried under 5s/p can minimize external influences and consequently enable precise and continuous tuning of Ru-O covalency for durable OER electrocatalysis."

Comment 2: The results in Fig. 1b-c are not consistent with the description in the manuscript. In Fig. 1b-c, the -ICOHP of Ho is larger than that of Tm, but in the main text, the -ICOHP of Ho is smaller (1.619 vs. 1.622). Unfortunately, the volcano trend would no longer exist.

Response 2: Thank you for your valuable comment. There was an error in the labeling of the figures. We apologize for the mistake. The previous calculated -ICOHP is the average of overall Ru-O bond in the model. To better describe the properties of the active sites, the values of -ICOHP has been changed from the average of overall Ru-O bond to specific active site in the revised manuscript. The detail explanation is provided in **Response 4**. The corresponding description and figures have been revised.

Revision: Line 101-104. "Through analysis of bonding and antibonding orbital filling, the crystal orbital Hamilton population (COHP) and integrated COHP (ICOHP) calculations demonstrate that the introduction of Ln can weaken the Ru-O bonding state occupancy from -1.614 eV (RuO₂) to -1.523

eV (Ho-RuO_x), -1.573 eV (Er-RuO_x) and -1.574 eV (Tm-RuO_x) (Supplementary Fig. 6).”

Supplementary Fig. 6 | Calculated COHP of the as-prepared catalysts. The COHP diagram of (a) Ho-RuO_x, (b) Er-RuO_x, (c) Tm-RuO_x and (d) RuO₂, respectively.

Fig. 1 | Prediction of the OER performance utilizing DFT calculations. (a) The qualitative

molecular orbital diagram obtained from [RuO₆] and [LnO₆] with O_h symmetry. (b) The $\Delta G_{\text{O vacancy}}$ and (c) $\Delta G_{\text{Ru vacancy}}$ as a function of -ICOHP for Ln-RuO_x. The upshift values of ICOHP indicates lower Ru-O covalency. (d) The reaction paths on Er-RuO_x and RuO₂ at 1.23 V. (e) Volcano plot for different electrocatalysts and corresponding structures.

Revision: Line 107-115. “To evaluate the stability of lattice oxygen and Ru, we calculated the formation energy of the lattice oxygen ($\Delta G_{\text{O vacancy}}$) and Ru vacancy ($\Delta G_{\text{Ru vacancy}}$), which are utilized together to assess the stability of the electrocatalysts. As Ln dissolution would take place during OER process³³, defective structures containing Ln vacancies were constructed. In the presence of Ln vacancies, Er-RuO_x exhibits the highest $\Delta G_{\text{O vacancy}}$ (0.33 eV), surpassing RuO₂, Ho-RuO_x and Tm-RuO_x by 0.29, 0.02 and 0.12 eV, respectively (Fig. 1b). Specifically, the regulation of Ru-O covalency leads to a modified $\Delta G_{\text{Ru vacancy}}$, increasing from 2.58 eV in RuO₂ to 3.49, 3.78 and 3.44 eV for Ho-RuO_x, Er-RuO_x and Tm-RuO_x, respectively (Fig. 1c). Considering the $\Delta G_{\text{O vacancy}}$ and $\Delta G_{\text{Ru vacancy}}$ in combination, the stability of Ln-RuO_x follows the volcanic-like trend as a function of Ru-O covalency.”

Comment 3: Why choose Tm/Er/Ho? The changes between these three elements are so small that they raise concerns about error variations in the results, such as 1.52/1.50/1.48 (Bader charge), -1.622/-1.636/-1.619 (ICOHP), 0.54/0.58/0.60 ($\Delta G_{\text{O vacancy}}$), and 2.49/2.48/2.57 ($\Delta G_{\text{Ru vacancy}}$).

Response 3: Thanks very much for the valuable comments. The reasons are as follows. Firstly, transition metals with d-valence electrons which are doped into or alloyed with RuO_x, are inherently susceptible to the influence of crystal field and coordination environment, making it challenging to precisely and continuously modulate the Ru-O covalency within a narrow range.

Secondly, as lanthanide-group elements with the 4f orbital buried under 5s/p could minimize external influences, we reason that introduction of Ln enable precise tuning of Ru-O covalency for durable OER electrocatalysis.

Thirdly, lanthanide-group elements exhibit biperiodic chemical trends, with Gd serving as the dividing line, where La to Gd form one peak and Gd to Lu form another peak. This pronounced division into two groups arises from the difference in the 4f electron occupation. In order to achieve monotonic modulation of Ru-O covalency, we selected the elements exhibiting linear variation in their chemical properties for investigation, namely Ho, Er and Tm.

Comment 4: Furthermore, in comparison of Bader charges (Tm>Er>Ho) and -ICOHP (Er>Ho>Tm), the trends are not the same (whether in ascending or descending order), which further makes the small fluctuations appear to be due to calculation error rather than a fine-tuning result. In principle, the ICOHP is mainly influenced by the degrees of electron occupancy when considering a set of similar models, which would be expected to match the amount of electrons transferred, but the following manuscript concentrates only on the ICOHP results, where possible reasons for the different trends are not discussed.

Response 4: Thanks very much for the valuable comments. The difference in the trends of Bader charge (Tm>Er>Ho) and -ICOHP (Er>Ho>Tm) arises from the calculation of different sites: Bader

charges represent the charge at active sites, whereas -ICOHP is the average of overall Ru-O bond in the model. To better describe the properties of the active sites, the values of -ICOHP has been changed from the average of overall Ru-O bond to specific active site in the revised manuscript, which is matched with the trends of Bader charge.

Revision: Line 101-104. “Through analysis of bonding and antibonding orbital filling, the crystal orbital Hamilton population (COHP) and integrated COHP (ICOHP) calculations demonstrate that the introduction of Ln can weaken the Ru-O bonding state occupancy from -1.614 eV (RuO₂) to -1.523 eV (Ho-RuO_x), -1.573 eV (Er-RuO_x) and -1.574 eV (Tm-RuO_x) (Supplementary Fig. 6).”

Supplementary Fig. 6 | Calculated COHP of the as-prepared catalysts. The COHP diagram of (a) Ho-RuO_x, (b) Er-RuO_x, (c) Tm-RuO_x and (d) RuO₂, respectively.

Revision: Line 107-115. “To evaluate the stability of lattice oxygen and Ru, we calculated the formation energy of the lattice oxygen ($\Delta G_{\text{O vacancy}}$) and Ru vacancy ($\Delta G_{\text{Ru vacancy}}$), which are utilized together to assess the stability of the electrocatalysts. As Ln dissolution would take place during OER process³³, defective structures containing Ln vacancies were constructed. In the presence of Ln vacancies, Er-RuO_x exhibits the highest $\Delta G_{\text{O vacancy}}$ (0.33 eV), surpassing RuO₂, Ho-RuO_x and Tm-RuO_x by 0.29, 0.02 and 0.12 eV, respectively (Fig. 1b). Specifically, the regulation of Ru-O covalency leads to a modified $\Delta G_{\text{Ru vacancy}}$, increasing from 2.58 eV in RuO₂ to 3.49, 3.78 and 3.44 eV for Ho-RuO_x, Er-RuO_x and Tm-RuO_x, respectively (Fig. 1c). Considering the $\Delta G_{\text{O vacancy}}$ and $\Delta G_{\text{Ru vacancy}}$ in combination, the stability of Ln-RuO_x follows the volcanic-like trend as a function of Ru-O covalency.”

Fig. 1| Prediction of the OER performance utilizing DFT calculations. (a) The qualitative molecular orbital diagram obtained from [RuO₆] and [LnO₆] with O_h symmetry. (b) The $\Delta G_{\text{O vacancy}}$ and (c) $\Delta G_{\text{Ru vacancy}}$ as a function of -ICOHP for Ln-RuO_x. The upshift values of ICOHP indicates lower Ru-O covalency. (d) The reaction paths on Er-RuO_x and RuO₂ at 1.23 V. (e) Volcano plot for different electrocatalysts and corresponding structures.

Comment 5: The trend of Ru vacancy formation energies is reversed before and after Ln leaching, with $\Delta G_{\text{Ru vacancy}}$ (Er<Ho<Tm) and $\Delta G_{\text{Ru vacancy}}$ after Ln-leaching (Er>Ho>Tm), so what about $\Delta G_{\text{O vacancy}}$? Would it also be reversed? If so, does it mean that the stability of O and Ru cannot be achieved simultaneously? At present, the authors chose $\Delta G_{\text{O vacancy}}$ before Ln leaching (Fig. 1b) and $\Delta G_{\text{Ru vacancy}}$ after Ln leaching (Fig.1c) to conclude, but the formation energies of these two kinds of vacancies should be compared under the same model in principle.

Response 5: Thanks very much for the valuable comments. According to your nice suggestion, the $\Delta G_{\text{O vacancy}}$ after Ln leaching was calculated, and the formation energies of two kinds of vacancies were compared under the same model. The values of $\Delta G_{\text{O vacancy}}$ after Ln leaching changed, and Er-RuO_x still exhibited the highest $\Delta G_{\text{O vacancy}}$. The corresponding expressions and Figure have been modified

in the revised manuscript.

Revision: Line 107-115. “To evaluate the stability of lattice oxygen and Ru, we calculated the formation energy of the lattice oxygen ($\Delta G_{\text{O vacancy}}$) and Ru vacancy ($\Delta G_{\text{Ru vacancy}}$), which are utilized together to assess the stability of the electrocatalysts. As Ln dissolution would take place during OER process³³, defective structures containing Ln vacancies were constructed. In the presence of Ln vacancies, Er-RuO_x exhibits the highest $\Delta G_{\text{O vacancy}}$ (0.33 eV), surpassing RuO₂, Ho-RuO_x and Tm-RuO_x by 0.29, 0.02 and 0.12 eV, respectively (Fig. 1b). Specifically, the regulation of Ru-O covalency leads to a modified $\Delta G_{\text{Ru vacancy}}$, increasing from 2.58 eV in RuO₂ to 3.49, 3.78 and 3.44 eV for Ho-RuO_x, Er-RuO_x and Tm-RuO_x, respectively (Fig. 1c). Considering the $\Delta G_{\text{O vacancy}}$ and $\Delta G_{\text{Ru vacancy}}$ in combination, the stability of Ln-RuO_x follows the volcanic-like trend as a function of Ru-O covalency.”

Fig. 1 | Prediction of the OER performance utilizing DFT calculations. (a) The qualitative molecular orbital diagram obtained from [RuO₆] and [LnO₆] with O_h symmetry. (b) The $\Delta G_{\text{O vacancy}}$ and (c) $\Delta G_{\text{Ru vacancy}}$ as a function of -ICOHP for Ln-RuO_x. The upshift values of ICOHP indicates lower Ru-O covalency. (d) The reaction paths on Er-RuO_x and RuO₂ at 1.23 V. (e) Volcano plot for different electrocatalysts and corresponding structures.

Comment 6: Please specify the atom sites which are considered in these calculations. Are these results from a single site or averaged over a number of sites? Are the atoms in the surface layer, sublayer, or full slab?

Response 6: Thank you for your valuable comment. The detailed information of atom sites has been specified in the revised manuscript.

Revision: Line 349-357. “RuO₂(110) surface (Supplementary Fig. 1) was modeled by a 2×2×2 supercell (with 55 O atoms and 24 Ru atoms), in which the top three atomic layers are allowed to relax. As for Ln-RuO_x, two Ru atoms in the RuO₂(110) model was substituted by Ln atoms (Supplementary Figs. 2-4), leading to a chemical composition of 8.3 at percentage Ln and 91.7 at percentage Ru. A vacuum spacing of 20 Å was set along the z-direction to prevent the interaction between the slab and its periodic motif. The termination of RuO₂(110) and Ln-RuO_x has all Ru filled with oxygen but one coordinatively unsaturated site Ru unfilled, which is the active site for intermediates adsorption. The Monkhorst-Pack method was used for sampling the Brillouin zone with a 3×3×1 mesh.”

Line 361-365. “When calculating the formation energy of Ru vacancies, the lost Ru atoms originate from unsaturated surface coordination sites, consistently from the same position across all models. In the calculation of oxygen vacancy formation energy, both Ln-RuO_x and RuO₂(110) lose oxygen atoms surrounding unsaturated Ru coordination sites on the surface. ICOHP and Bader charge represent the values at active site.”

Comment 7: Care should be taken when using expressions such as “the most stable catalyst”. From the Supplementary Tables 1 and 2, it seems that there is no significant difference in the performances between Er-RuO_x and Ni-RuO_x (Nat. Mater. 2023,22, 100-108), except for a 14 mV decrease in η for the latter. Given that Ni is much more abundant than Er, what are the advantages of using Er?

Response 7: Thank you for your valuable comment. According to the suggestion, we have revised the corresponding expressions in the manuscript.

Revision: Line 23-26. “Most importantly, the Er-RuO_x-based PEMWE requires only 1.837 V to reach 3 A cm⁻² and shows a long-term stability at 500 mA cm⁻² for 100 h with a degradation rate of mere 37 μ V h⁻¹, making the Er-RuO_x be one of the most stable RuO_x-based catalysts so far.”

Ni-RuO₂ is a promising catalyst, demonstrating high catalytic activity and stability in acidic OER for PEM water electrolysis. Each of these works has its merits. The advantages of Er-RuO_x are as follows:

(1) Er-RuO_x possesses the porous sheet-like structure, which reduces mass and charge transport resistance in the PEMWE device. Specifically, the Er-RuO_x-based PEMWE device requires only 1.59 V to reach an industrial current density of 1 A cm⁻², outperforming the Ni-RuO₂-based PEMWE (1.95 V @ 1 A cm⁻², Supplementary Table 8).

(2) Er-RuO_x exhibits superior stability compared to Ni-RuO₂. Specifically, the stability of both Er-RuO_x and Ni-RuO₂-based PEMWEs have been tested at 200 mA cm⁻², and no significant activity decay

was observed after long-term stability test for them. Additionally, the Er-RuO_x-based PEMWE electrolyzer exhibits good stability at a high current density of 1 A cm⁻², while Ni-RuO₂ does not demonstrate stability at such high current densities.

(3) Lanthanide-group elements with the 4f orbital buried under 5s/p could minimize external influences and consequently enable precise and continuous tuning of Ru-O covalency for durable OER electrocatalysis.

Comment 8: Considering the possible different synthesis/characterization procedures, it might be more convincing to include the control experiments of d-element doping, such as Ni. Also, if one wants to emphasize continuous tuning when using f-elements to dope RuO_x, it would be better to include control experiments with different f-element doped samples.

Response 8: Thank you for your valuable comments. The reason why we do not choose the transition metals with d-valence electrons doping into RuO₂ has been explained in the part of **Response to Comment 3**.

According to your nice suggestion, Ho-RuO_x and Tm-RuO_x was synthesized *via* the same method for comparison. The characterization and OER performance of Ho-RuO_x and Tm-RuO_x was provided in the Supplementary Information. The element mapping images (Supplementary Figs. 19-20) demonstrate that Ho and Tm was successfully introduced into RuO₂. The OER activity and stability of Ho-RuO_x and Tm-RuO_x (Supplementary Figs. 21-23) was in good agreement with DFT prediction.

Supplementary Fig. 19 | TEM characterization of Ho-RuO_x. (a) TEM and (b) HRTEM images of Ho-RuO_x. (c) HAADF-STEM image and (d-f) corresponding element mapping images of Ho-RuO_x.

Supplementary Fig. 20 | TEM characterization of Tm-RuO_x. (a) TEM and (b) HRTEM images of Tm-RuO_x. (c) HAADF-STEM image and (d-f) corresponding element mapping images of Tm-RuO_x.

Supplementary Fig. 21 | OER polarization curves of Ho-RuO_x, Er-RuO_x and Tm-RuO_x.

Supplementary Fig. 22 | The CP curves of Ho-RuO_x at 10 mA cm⁻².

Supplementary Fig. 23 | The CP curves of Tm-RuO_x at 10 mA cm⁻².

Revision: Line 187-188. “The OER performance of Ho-RuO_x and Tm-RuO_x was shown in Supplementary Figs. 21-23, being in good agreement with DFT prediction.”

Comment 9: To obtain information about the occupancy of bonding/antibonding states, the results of COHP between *OH and the adsorption site (Ru) could provide more direct evidence compared to the dz² band center.

Response 9: Thank you for your valuable comment. According to the suggestion, the COHP between *OH and the adsorption site (Ru) was calculated.

Supplementary Fig. 9 | Calculated COHP and ICOHP for Ru-O between *OH and catalysts: (a) RuO₂, (b) Er-RuO_x.

Revision: Supplementary Information

“COHP and ICOHP calculations (Supplementary Fig. 9) demonstrated that introduction of Er could obviously increase the Ru-*OH bonding state occupancy, contributing to a higher OER activity.”

To Reviewer 4:

Overall comments: The authors used lanthanide-group (Ln) elements to regulate ruthenium (Ru)-oxygen (O) covalency to enhance the electrocatalytic stability of RuO_x. The authors claimed the 4d-2p-4f orbital hybridization resulted in the improved both oxygen evolution reaction (OER) activity and stability. This work is interesting and may attracts community attention. Nevertheless, some revisions are required prior to further consider in Nature Communications.

Response: We appreciate your constructive comments, which have helped us to substantially improve the quality of manuscript. We have revised the manuscript accordingly.

Comment 1: Since the formation energy trends of RuO_x doped with Ho, Tm, and Er in Figure 1b and c are different, the additional discussion is necessary for the variation.

Response 1: Thank you for your valuable comment. The trends in $\Delta G_{O \text{ vacancy}}$ and $\Delta G_{Ru \text{ vacancy}}$ for Ln-RuO_x are similar, both increasing initially with the increase in -ICOHP, followed by a decrease. Among them, Er-RuO_x exhibits the highest $\Delta G_{O \text{ vacancy}}$ and $\Delta G_{Ru \text{ vacancy}}$, theoretically suggesting its optimal stability. Detailed explanations are provided in the revised manuscript.

Revision: Line 107-115. “To evaluate the stability of lattice oxygen and Ru, we calculated the formation energy of the lattice oxygen ($\Delta G_{O \text{ vacancy}}$) and Ru vacancy ($\Delta G_{Ru \text{ vacancy}}$), which are utilized together to assess the stability of the electrocatalysts. As Ln dissolution would take place during OER process³³, defective structures containing Ln vacancies were constructed. In the presence of Ln vacancies, Er-RuO_x exhibits the highest $\Delta G_{O \text{ vacancy}}$ (0.33 eV), surpassing RuO₂, Ho-RuO_x and Tm-RuO_x by 0.29, 0.02 and 0.12 eV, respectively (Fig. 1b). Specifically, the regulation of Ru-O covalency leads to a modified $\Delta G_{Ru \text{ vacancy}}$, increasing from 2.58 eV in RuO₂ to 3.49, 3.78 and 3.44 eV for Ho-RuO_x, Er-RuO_x and Tm-RuO_x, respectively (Fig. 1c). Considering the $\Delta G_{O \text{ vacancy}}$ and $\Delta G_{Ru \text{ vacancy}}$ in combination, the stability of Ln-RuO_x follows the volcanic-like trend as a function of Ru-O covalency.”

Comment 2: It is necessary to explain why only Ho, Er, and Tm were chosen for the calculation and comparison among various Lanthanide-group elements.

Response 2: Thank you for your valuable comment. The reasons for choose of Tm/Er/Ho are as follows. Lanthanide-group elements exhibit biperiodic chemical trends, with Gd serving as the dividing line, where La to Gd form one peak and Gd to Lu form another peak. This pronounced division into two groups arises from the difference in the 4f electron occupation. In order to achieve precise and monotonic modulation of Ru-O covalency, we selected the elements exhibiting linear variation in their chemical properties for investigation, namely Ho, Er and Tm.

Comment 3: While Er-RuO_x theoretically expected to exhibit the best activity and stability, and it was experimentally validated. However, Ho-RuO_x and Tm-RuO_x should also be additionally synthesized to experimentally demonstrate that their OER activity and stability are superior to RuO₂ but inferior to Er-RuO_x.

Response 3: Thank you for your valuable suggestion. According to the suggestion, Ho-RuO_x and Tm-RuO_x were synthesized by the same method. The OER performance of Ho-RuO_x and Tm-RuO_x have been added to the Supplementary Information, which is in good agreement with DFT prediction.

Supplementary Fig. 21| OER polarization curves of Ho-RuO_x, Er-RuO_x and Tm-RuO_x.

Supplementary Fig. 22| The CP curves of Ho-RuO_x at 10 mA cm⁻².

Supplementary Fig. 23| The CP curves of Tm-RuO_x at 10 mA cm⁻².

Revision: Line 187-188. “The OER performance of Ho-RuO_x and Tm-RuO_x was shown in Supplementary Figs. 21-23, which was in good agreement with DFT prediction.”

Comment 4: It would be beneficial to include control experiments to determine the optimum Er ratio.

Response 4: Thank you for your valuable suggestion. The OER performance of Er_{0.05}-RuO_x, Er_{0.1}-RuO_x, Er_{0.2}-RuO_x were measured in 0.5 M H₂SO₄ solution in a three-electrode system. The corresponding figure have been added to the Supplementary Information.

Supplementary Fig. 17| OER polarization curves of Er_{0.05}-RuO_x, Er_{0.1}-RuO_x and Er_{0.2}-RuO_x. Among them, Er_{0.1}-RuO_x exhibits the best OER activity. Unless otherwise specified, Er-RuO_x mentioned in the manuscript refers to Er_{0.1}-RuO_x.

Comment 5: The claim of successfully achieving hydrogen generation at a lower cost according to the US Department of Energy (DOE) standards lacks explanation in this paper. Please add relevant data comparison.

Response 5: Thank you for your valuable comment. The calculation of the hydrogen production cost and corresponding description has been provided in the revised manuscript and Supplementary Information.

Revision: Supplementary Information

“Cost of per kilogram H₂^[1]:

Cost (H₂/kg) = energy consumption × electricity bill

$$= 42.61 \text{ kW h/kg H}_2 \times \$ 0.02/\text{kW h}$$

$$= \$ 0.85/\text{kg H}_2$$

1. Hao, S. et al. Torsion strained iridium oxide for efficient acidic water oxidation in proton exchange membrane electrolyzers. *Nat. Nanotechnol.* **16**, 1371-1377 (2021).”

Revision: Line 26-28. “The estimated hydrogen production cost (US\$ 0.85 *per* kg H₂) at 1 A cm⁻² over Er-RuO_x-based PEMWE is much lower than that of the U.S. Department of Energy’s target (US\$ 2 *per* kg H₂).”

Comment 6: It is recommended to organize the figures. Upon reviewing Supporting Figures 9 and 10, the colors of the samples in the supporting are different from the main text.

Response 6: Thank you for your valuable comment. Supplementary Figure 9 and 10 have been renamed to Supplementary Figure 7 and 8, respectively. Their colors have been revised in the supplementary information to match those in the main text.

Supplementary Fig. 7 | The reaction paths on Ho-RuO_x with the set potential of 1.23 V.

Supplementary Fig. 8 | The reaction paths on Tm-RuO_x with the set potential of 1.23 V.

Comment 7: Check and correct references and typos.

Response 7: Thank you for your valuable comment. We have checked and corrected the references and typos.

REVIEWERS' COMMENTS

Reviewer #1 (Remarks to the Author):

This revision is recommended for publication.

Reviewer #2 (Remarks to the Author):

The authors have addressed all my concerns well. now it can be accepted for publication.

Reviewer #3 (Remarks to the Author):

All the concerns are addressed well, and thus it is recommended to be published as it is.

Reviewer #4 (Remarks to the Author):

Having reviewed the revised version of this manuscript, the reviewer is pleased to see that the suggestions have been effectively implemented. The modifications have notably enhanced the clarity and coherence of the manuscript. Given the improvements made, the reviewer sees no objections for its publication at current form.